# Three-Class EEG-Based Motor Imagery Classification Using Phase-Space Reconstruction Technique

**DOI:** 10.3390/brainsci6030036

**Published:** 2016-08-23

**Authors:** Ridha Djemal, Ayad G. Bazyed, Kais Belwafi, Sofien Gannouni, Walid Kaaniche

**Affiliations:** 1EE Department, King Saud University, Riyadh 11421, Saudi Arabia; ayad.gaafar@gmail.com; 2CS Department, King Saud University, Riyadh 11421, Saudi Arabia; kbelwafi@ksu.edu.sa (K.B.); gnnsof@ksu.edu.sa (S.G.); 3Electrical Engineering Department, ENISo of Sousse, BP 264 Erriadh 4023, Sousse 4054, Tunisia; walidkaaniche@gmail.com

**Keywords:** brain-computer interface (BCI), motor imagery (MI), electroencephalogram EEG

## Abstract

Over the last few decades, brain signals have been significantly exploited for brain-computer interface (BCI) applications. In this paper, we study the extraction of features using event-related desynchronization/synchronization techniques to improve the classification accuracy for three-class motor imagery (MI) BCI. The classification approach is based on combining the features of the phase and amplitude of the brain signals using fast Fourier transform (FFT) and autoregressive (AR) modeling of the reconstructed phase space as well as the modification of the BCI parameters (trial length, trial frequency band, classification method). We report interesting results compared with those present in the literature by utilizing sequential forward floating selection (SFFS) and a multi-class linear discriminant analysis (LDA), our findings showed superior classification results, a classification accuracy of 86.06% and 93% for two BCI competition datasets, with respect to results from previous studies.

## 1. Introduction

In recent decades, brain-computer interfaces (BCIs) have been developed as a new type of human-computer interface (HCI) [1]. A BCI system aims to assist people who are suffering from severe movement disabilities to improve their interaction with the environment through an alternative interface, working as a communication channel between a brain and a computer [2]. A BCI allows the user to interact with other people and with external devices only by brain activity, such as thinking and imagining, without using any peripheral muscle activity. These mental activities lead to changes in the brain’s signals. A BCI system is responsible for acquiring, measuring, and converting these brain signals into control commands. This new type of HCI is useful, especially for people with motor problems. Previous studies presented different kinds of motor imagery-based BCI designs that require the integration of different algorithms. Choosing a group of specific algorithms is carried out according to different variables (e.g., positions of electrodes, number of EEG channels, number of states, application commands, validation methods), and the choice leads to different levels of classification accuracy.

Several three-class BCIs based on motor imagery (MI) have been developed [3,4,5,6,7,8,9,10]. In [3], authors extracted EEG features using fast Fourier transform (FFT). The accuracy reached 80%, and most classification errors were due to the third state (word), because the FFT could not distinguish between hands and word. Since EEG data are non-stationary signals, applying the Fourier transform alone is not sufficient to extract significant features from them [4]. Another study attempted to classify repetitive left hand, right hand, and word using the feature selector of power spectral density features called sequential forward floating selection (SFFS). The results showed that classification using a latent-dynamic conditional random field model achieved accuracy ranging between 69.5% and 85%, which is more accurate than that achieved using a conditional random field model (67%–81%) [5]. In [6], control sessions for humanoid robots were achieved by extracting the amplitude features using power spectral analysis, while most features were selected by Fisher’s discriminant ratio. The states were the left hand (LH), right hand (RH), and foot (F) MI of five healthy subjects. The quadratic discriminant analysis technique was efficiently used, providing accuracy close to 80% by using a switch to distinguish between the MI and non-MI features. However, this switching-based method is difficult and unsuitable in some applications, such as those in which the start and end of the trial are undetermined. In [7], the authors presented a three-class MI BCI system (LH, RH, and F) for controlling a simulated mobile robot. They attempted to increase the accuracy based on two modes, no control (NC) and in control (IC) modes, which were switched by a linear discriminant analysis (LDA) classifier to predict the MI. Then, another LDA classifier was applied to classify the features passed through the IC part. The classifiers were combined to control the switching between the NC and IC modes. The complexity of such an experiment is due to the distinction between MI and non-MI cases. Thus, the accuracy is increased accordingly, and subjects need more training sessions and should only imagine when the BCI system is online.

One of the most widely used techniques in BCIs for reducing the high-dimensionality issue is filtering using the common spatial pattern (CSP) algorithm [8]. The CSP algorithm is used to improve the signal-to-noise ratio (SNR) by projecting EEG signals related to two motor imageries to spatially convert a large amount of EEG data into a low-dimensional matrix containing the same information about the data [9]. Nevertheless, CSP is highly vulnerable to noise and outliers, leading to variable accuracy. To avoid this problem, the regularized CSP algorithm has been proposed to overcome the noise sensitivity problem of the CPS algorithm [10]. However, the eigenvectors do not always guarantee the discrimination of different tasks.

Several previous studies have proposed a combination of different features extracted in different domains [11,12,13,14,15,16]. In [11], the band power (BP) features of mu rhythms were extracted in a specific time window and followed by the CSP algorithm. The average classification accuracy of five subjects was 85%, and it increased to 88% by combining CSP with wavelet packet decomposition (WPD) features. The combination of some algorithms may be more time consuming, especially for the large number of EEG channels that represent the main problem in real-time systems. As for four-class BCI, most papers have presented results of the average classification accuracy being less than 75% with the MI tasks including the LH, RH, F (or both F), and tongue (T). In addition to the complexity of extracting the most features of four MI tasks, it is important to find a combination of suitable classifiers for distinguishing four different feature vectors.

The results in [12] showed the best classification accuracy of multivariate autoregressive (AR) and invariant AR models when using Burg’s algorithm to compute the AR coefficients. Several studies showed that neither FFT nor AR coefficients alone is enough to discriminate the EEG signals of multiple tasks [4]. Phase synchronization techniques have used amplitude and phase, as in [13], to solve some problems of BCI synchronization as a nonlinear method. The most widely used nonlinear method is phase-space reconstruction, which is considered a promising method for non-stationary EEG signals [14]. In [15], Fang used phase-space reconstruction to extract the features of two classes. They combined a large number of features for only three channels in many frequency bands without selecting the optimal features. This constitutes a problem for the classifier. In [16], Townsend made a comparison between CSP and BP containing the amplitude and phase and found that BP presented better results than CSP. However, the classification accuracy is still limited, perhaps due to the presence of artifacts and the fact that the feature extraction and classification require greater improvements. For these reasons, choosing the best feature is one of the challenging problems in EEG signal processing, and it should be done carefully as suggested in [17,18]. One of the main objectives of this paper is to provide an efficient feature extraction technique where the parameters are customized for each subject to improve the classification accuracy. Hence, we propose to enhance the extracted features from recorded EEG signals using FFT and AR algorithms to extract the amplitude and the phase. The large number of features obtained by this idea will bring about a problem related to the classification section. Therefore, we will combine the features in order to select the most useful information in a few features. We also propose some procedures to improve the classification accuracy by integrating a dedicated filtering technique and by adjusting the useful subband frequencies related to α and β rhythms.

During the first step, we extract the amplitude and the phase using FFT. Some features will be extracted by phase-space reconstruction and autoregression. Using this approach, we can convert the attractor of the system into another one that has the same dynamic properties by reconstructing the dimensions of the origin series. This procedure will help us to detect hidden features that can be sufficiently classified. The second features will be extracted by FFT to obtain amplitude and phase information as well. All features will be combined to enhance the variance among them, and a surface Laplacian filter will be used to improve the spatial resolution of channels as a preprocessing step. Moreover, since we need to reduce the computational burden without significantly affecting the performance of the classification system, and since we need to reach optimality and efficiency, we seek an algorithm based on a wrapper search method that adds and removes features sequentially. In the final step, we develop a multi-class LDA or support vector machine (SVM) classifier adopting the one-against-one strategy, which builds one LDA (or SVM) for each pair of classes. All classifiers try to classify the input vector to one class (LH, RH, or F) among the three classes available. The final decision is obtained using a majority voting technique. This step will solve some problems appearing at the multi-class classification step. The remainder of the paper is organized as follows: Section 2 describes the method used in our work, including data description, preprocessing, feature extraction, and classification methods. Experimental results are presented in Section 3, while Section 4 discusses the results and concludes the paper.

## 2. Materials and Methods

In this section, we describe the method of feature extraction and classification techniques as well as their validation based on MATLAB simulation. All possible combinations of the preselected approach were coded and tested. Furthermore, we processed the EEG signals using some optimization techniques (channel selection, feature selection, time of trial) to increase the accuracy of the approach. Various parameters were modified in order to analyze, extract, and reduce EEG data information in the time, frequency, or spatial domains. Therefore, we varied the values of the time window offset (time shift), frequency band of the band-pass filter, number of best features, and type of classification combination to improve the classification result while reducing the required processing time.

The main idea consisted of extracting the amplitude and the phase of EEG samples in order to obtain the best possible features. The optimal features were obtained by an appropriate selection method considered an input vector to the classifier. The classification results were obtained by three types of classifiers, two of which were built around a set of two-class LDA or SVM classifiers. Both SVM and LDA techniques consider a hyperplane to distinguish between two classes. However, they behave differently with regard to the classification of high dimensionality, and the required processing time differs significantly. The other classifier was a single multi-class classifier (K-nearest-neighbor KNN classifier) without any combination. Figure 1 shows the algorithm suggested to improve the accuracy of the three-class MI BCI.

### 2.1. Data Set Description

Two public datasets of a BCI competition, provided by Graz University of Technology, were used in our experiment. These datasets were recorded when the subjects performed four different types of MI tasks (LH, RH, F, and T) [19,20].These datasets were organized as follows: Dataset IIa [20], from BCI competition IV, consisted of EEG data acquired from nine subjects performing four different MI tasks (i.e., LH, RH, F, and T). The data were recorded in two different sessions using 25 electrodes where three of them contained EOG artifacts. EEG signals were sampled with 250 Hz and filtered between 0.5 and 100 Hz. The recorded data for each subject contained 288 trials.Dataset IVa [21], from BCI competition III, contained EEG signals from three subjects integrating four different MI tasks (i.e., LH, RH, F, and T). The data were acquired through 60 electrodes sampled with 250 Hz and filtered between 1 and 50 Hz. The recorded data contained 80 trials for each class.


### 2.2. Preprocessing

As mentioned above, a large number of methods have been introduced to EEG analysis for MI applications using band-pass filters [22]. However, to guarantee a higher level of accuracy for MI classification, we explored preprocessing techniques, including both spatial and band-pass filter combinations with time window and offset regulation, in depth.

Although the power of the alpha and beta frequency bands of a training dataset can be discriminated during MI at the channels (C3, Cz, and C4), the MI features are not balanced and can be changed depending on the subject [4]. To solve this problem, we used a surface Laplacian filter [23]. The surface Laplacian filter improves the spatial resolution of the C3, Cz, and C4 electrodes by estimating the surrounding potentials of these electrodes. This procedure can be done by derivating the approximate potentials along certain surfaces. Using this filter, we can control the amount of spatial smoothing on EEG signals based on the electrodes’ positions. In our experiments, 15 channels were spatially filtered to generate linear combinations of the electrodes in order to produce three useful channels containing the maximum amount of contributive information with the least noise possible.

EEG signals contain artifacts resulting from eye or body movements. We can get rid of the artifacts by rejecting trials contaminated by artifacts (e.g., using a specified threshold); however, this procedure can be unsuitable and can even have an effect on classification accuracy due to our need for all trials of the training set. The dataset of Graz contained fewer eye artifacts, because three EOG electrodes were used during the recording. In addition, the subjects were trained to fix their eyes without blinking, but there were still some errors during data recording. Eye artifacts occur on a frequency band between 0 and 4 Hz, and the ERD/ERS phenomenon occurs in the mu and beta frequency bands (8–32 Hz) [24]. Thus, we used the elliptic band-pass filter, because it provides better experimental accuracy compared with other filters, such as Chebyshev type I and type II and Betterworth, which are IIR filters [22]. Furthermore, the implementation of the elliptic filter requires less memory and calculation and provides reduced time delays compared with all other FIR and IIR filtering techniques. The MI is not located exactly along the mu and beta rhythms but varies depending on the subject. Moreover, extracting the features from many bands leads to the accumulation of more information in a massive vector, which may cause a problem for the classifier. To solve this problem, we divided the band into two groups: 8–18 Hz and 18–32 Hz. Several time windows of the trials with various offsets were taken to extract the most useful features from all subjects and to avoid the high dimensionality that could have led to inaccurate results. Time windows and offsets will be discussed in more detail in Section 3.

### 2.3. Feature Extraction

Traditionally, in BP feature extraction, EEG samples are squared and averaged, and then the logarithms of values are taken. This approach is rudimentary and undeveloped, as the information is insufficient, especially in the case of multi-class tasks [25]. In order to enhance the feature analysis, we proposed extracting the phase information as well as the amplitude of the EEG samples according to the following techniques.

#### 2.3.1. Amplitude-Phase Features

FFT was used for each channel to perform the discrete Fourier transform computation and extract the amplitude and the phase in an efficient manner [26]. Since the EEG signal does not have a convenient closed form of the FFT transforms, as it is non-periodic, its frequency spectrum suffers from non-zero values of signal energy spreading over frequency range sidelobes (commonly called leakage). To reduce the leakage, the FFT can be applied with a finite time window of the signal [27]. Preliminary results without using phase reconstruction are presented in [28] providing a limited accuracy rate from 58% to 75%.

In our experiment, the FFT was applied to the product of the EEG signal samples and a sliding time window using the Hamming function. This Hamming window should be short enough to reduce the leakage. The length of the segment used by the FFT was set to 64 samples wide at the sampling rate of 250 Hz used for the recordings (64/250 = 0.256 s). The Hamming function is also useful to improve the SNR by uniformly distributing the noise while collecting most of the signal energy around one frequency. The FFT results are complex coefficients (with real and imaginary parts) from which the amplitude and phase information are extracted, as denoted in (1) and (2). The amplitude is taken by calculating the absolute of the real value, and the phase is taken by calculating the angle of the complex vector.
(1)α(t,f)=|X(t,f)|           f∈S(f)
(2)φ(t,f)=tan−1Im(X(t,f))Re(X(t,f))
where *X*(*t*, *f*) is the complex coefficient resulting from the FFT, *α*(*t*, *f*) is the amplitude, φ(*t*, *f*) is the phase angle, and *S*(*f*) is the frequency spectrum of the EEG signal. In the final step, the features of phase and amplitude are averaged for each channel and collected in one vector. These features are then combined with the AR features into a single vector.

#### 2.3.2. The Combined Phase-Space Reconstruction and AR Model Technique

The phase-space reconstruction technique is used to estimate the near-optimal parameters of the embedding dimension and delay time while converting the attractor of the system into another attractor having the same dynamic properties by reconstructing the dimensions of the original series. This technique is applied for chaotic series to increase the reconstruction accuracy without increasing the computational complexity [29,30]. In fact, the complexity of the proposed technique is relatively lower in compare with that of conventional techniques and not strongly dependent on the data length. Thus, we apply this technique for EEG signal feature extraction. It is very important to select a suitable pair of parameters represented by the dimension m and the time delay τ when performing phase-space reconstruction. In our case, due to the presence of many artifacts in the EEG signal and the fact that the times series in the real word is not infinitely long, m and *τ* are strongly correlated. Consequently, the time window is defined as follows: $t_{W}=(m-1)\tau$. The phase-space reconstruction technique is able to detect hidden features that can be sufficiently classified. At first, we reconstructed all the dimensions of the signals using the time-delay phase-space reconstruction method, which is suitable for 1D time series, as it does not require that the system be mathematically defined. In such a method, the phase space of m-dimensions of the time series: {*x_i_*: i = 1, 2, …, N} can be represented by:
(3)$$Xi=x_{i},\;x_{i+\tau },\;\ldots ,\;x_{i+(m-1)\tau }$$ where M = N − (m − 1) *τ*, M is the number of data points used for the estimation, *τ* is the time delay, and m is the embedding dimension of our system. The subscript i goes from 1 to M = N − (m − 1) *τ*. The reconstructed vector *X* can also be represented by:
(4)$$X=\left[\begin{array}{l} X_{1}\\ X_{2}\\ \;\vdots \\ X_{M} \end{array}\right]=\left[\begin{array}{l} x_{1}\;x_{1+\tau }\;\ldots \;x_{1+(m-1)\tau }\\ x_{2}\;x_{2+\tau }\;\ldots \;x_{2+(m-1)\tau }\\ \vdots \\ x_{M}\;x_{M+\tau }\;\ldots \;x_{N} \end{array}\right]$$

Such settings provide a precise description of the dynamics of the system when m is properly selected, because the obtained m-dimensional space acts as a pseudo state space.
(5)$$Xi=x_{i},\;x_{i+\tau },\;\ldots ,\;X_{i+(m-1)\tau }$$

The time delay n*τ* is an exponential function in the frequency domain: (6)$$x_{n}(t)-x(t+n\tau )\;\to \;X_{n}(\omega )\;=\;X(\omega ).e^{j\omega n\tau }$$

Hence, the phase-space reconstruction can be expressed as: (7)$$X_{n}(\omega )\;=\;X(\omega )\times F_{n}(\omega )$$ where *X*(*ω*) is the Fourier transform of *x*(*t*) and *F_n_*(*ω*) is the Hamming window. Hence, the reconstruction of the phase space of each dimension can be considered as a filter for the time series.

In the second step, we modeled the EEG signal at each channel using the AR method. Computationally, AR assumes the EEG signal as a linear combination of the signals at past time points, which provides an efficient representation of EEG signals [31]. We used the AR model, as it introduces parameters unaffected by the changes related to the subject, such as skull thickness. Mathematically, the EEG single channel *y*(*t*) can be modeled as an AR model of order *p*:
(8)$$y(t)=\sum_{i=1}^{p}\alpha_{i}\times y(t-i)+\varepsilon_{t}$$ where *p* is the number of past points (orders) that are used to model the current time point, *α_i_* (*i* = *1, 2*, ..., *p*) are the AR coefficients, and *ε_t_* is the zero-mean process. The coefficients of each dimension of each channel were averaged to obtain one feature for each dimension, which have desirable properties for the subsequent analysis. The final feature extraction step was to collect all the features obtained from both the amplitude—phase and AR modules to serve as input vectors to the classifier. However, the large number of features usually leads to a failure in the separation of states [4]. Thus, the optimal features were selected using a feature selection algorithm. All features resulting from the AR-based feature extraction operation were subjected to the cross-validation process, including how the training and test subsets were classified for one subject. These features were split into training and testing subsets containing class states: (k-1) subset for training and one for testing. In the training phase, the classifier was trained with training trials nine times each time one vector was transmitted into the testing classifier. Each result of the testing classifier was compared to the states of test trials for validation. This procedure was repeated nine times. All the results were averaged to produce only one classification accuracy score for each subject.

### 2.4. Feature Selection

The main goal of this stage is to select an optimal feature subset of BP features and avoid the problem of the dimensionality in the feature space. This step reduces the computational burden without significantly affecting the performance of the classification system. Since we need to realize optimality and efficiency, we seek an algorithm based on the wrapper search method that adds and removes features sequentially. This algorithm is called the SFFS method [32]. Thanks to this method, we can solve the nesting problem, in which the feature selected in a step of the iterative process cannot be excluded in a later step. Since we use the voting of three classifiers (see Section 2.5), we propose three modules of SFFS, each of which selects the best subset of the features. In other words, one module of the classifier takes the best subset from one SFFS module. In the training, we collect the features of each module of the classifier together (RH vs. LH, RH vs. F, and LH vs. F), and then, the best features are applied to the test data. In this case, the two-class feature selection method produces more discriminative features in order to be correctly classified. This method is resorted to only when using multi-LDA or multi-SVM. SFFS needs more time to select the best subset, but there is no need to significantly reduce computational time in the training step as in our research. For the test data, the selected features are directly applied with no need for SFFS. Therefore, SFFS techniques are interesting, as they seem to provide an optimal classification solution.

In SFFS, *Y_k_* is considered the best selected subset of set (*X*), which begins as an empty set (*Y*_0_). The best feature x+ included in *X* is added to *Y_k_* when improving the results of the classification rate *J*(*Y_k_* + x^+^). Then, the worst feature *x*- is searched for to be eliminated from *Y_k_* when increasing the classification rate *J*(*Y_k_* − x^−^). This iteration takes place until the classification rate does not increase. The algorithm is described as following:
*Begin: set*
$Y_{0}=\{0\}$   *L1: while*
$x^{+}=\arg\max_{x^{+}\in Y_{k}}[J(Y_{k}+x^{+})]$
*—select the next best feature*     *If *
$J(Y_{k}+x^{+})>J(Y_{k})$   $Y_{k+1}=Y_{k}+x^{+};\;k=k+1$     *End if*   *L2: while*
$x^{-}=\arg\max_{x^{-}\in Y_{k}}[J(Y_{k}-x^{-})]$
*—remove the worst feature*     *If*
$J(Y_{k}-x^{-})>J(Y_{k})$   $Y_{k+1}=Y_{k}-x^{-};\;k=k+1$     *End if**End*


The cross-validation is a validation technique that can be used to estimate the performance of classifier. We used k-fold cross validation in all our experiments. In k-fold cross-validation, the dataset is randomly divided into k equal parts (k subsets). All the subsets are used to the training except one for the test (validation). The cross-validation is repeated k times (folds), then the results of k times are averaged to produce a single classification rate. In our case, each one of three classifiers was given (216 trials of IIa or 135 trials of IVa) trials, (72 or 45) trials of each class, 10-fold cross validation is recommended in machine learning. We applied 9 fold cross to avoid the remainder when splitting the trials subsets (135/10 = 13.5 subsets). Therefore, for each validation term, 108 or 192 trials have been used for training where 15 or 24 trials are applied for testing.

### 2.5. Classification

Three kinds of classifiers were used to classify the extracted features: LDA, SVM, and KNN. Since both LDA and SVM use a hyperplane to discriminate between classes, it is difficult to discriminate between three classes. Therefore, we combined three modules of a two-class LDA (or SVM) classifier to develop a three-class classifier. Each one translates the input vector, which is the same for all classifiers, to one class (LH, RH, or F). The final decision is obtained by a majority voting algorithm, as illustrated in Figure 2. We specified the signs (+ or −) as a result of each classifier: a positive sign for “LH MI” in classifiers 1 and 2, a negative sign for “F MI” in classifiers 2 and 3, and a negative sign and a positive sign for “RH MI” in classifiers 1 and 3, respectively.

Table 1 illustrates the majority voting process for the three MI states. For example, if the first classifier provides an LH signal as a result, where the second one gives an LH signal as the output of the classifier, the final result will be considered an LH MI regardless of the result of the third classification (represented by the symbol ∀ to indicate that it does not affect the vote output). The proposed organization represents a simple implementation of the so-called multi-class SVM (multi-SVM) and multi-class LDA (multi-LDA) classifiers, and it seems to be more suitable for embedded system integration compared with other multi-class implementations (e.g., Gaussian process classification [33]) because of its simplicity and its reduced processing time for the training and testing phases.

In our case, each classifier module was given 216 trials of the IIa dataset or 135 trials related to the IVa dataset. Ten-fold cross-validation is recommended in machine learning [34]. We applied nine-fold cross-validation to avoid a remainder when splitting the trial subsets (135/10 = 13.5 subsets). Therefore, for each validation term, 108 or 192 trials were used for training, whereas only 15 or 24 trials were considered for testing. The classification accuracy was calculated for each subset, and then the results were averaged to evaluate the performance of the algorithm. The classification accuracy was defined as: (9)$$Accuracy=\left(\frac{N_{correct}}{N_{total}}\right)\times 100\%$$ where *N_total_* was the number of overall samples to be classified and *N_correct_* was the number of correct samples. In order to compare our results with BCI competition results, we calculated the kappa score [35]. The kappa measure ranges between 1 and −1, where 1 indicates the best correct classification and −1 indicates the worst. It can be defined as: (10)$$k=\frac{\mathrm{Pr}(a)-\mathrm{Pr}(e)}{1-\mathrm{Pr}(e)}$$ where Pr(*a*) is the observed accuracy and Pr(*e*) is the probability of one class of observed data being correct.

## 3. Results

We present the results obtained for the proposed classification algorithm. The classification accuracy obtained from the test datasets is reported for three classifiers: LDA, SVM, and KNN. Then, the results of the best approach are presented with proper variations (frequency band of filter, size of time window, feature selection, and type of classification combination). To implement our algorithm, we used Matlab-R2013a.

In the first step of the feature extraction algorithm, we implemented the phase-space reconstruction with KNN. Table 2 shows the classification accuracy obtained by the approach for all subjects on the test set for nine-fold cross-validation. There were clear differences between subjects’ rates. Subject 1 had the best result, while subject 6 had the worst result. The average classification accuracy for the nine subjects was 80.71%. The average processing time of the algorithm was reported starting from the loading of the signal until the final decision. The total processing time for the training step of one subject was about 3 minutes, running on an Intel processor with a speed of 2.20 GHz.

As there was no improvement in terms of accuracy compared to published results, we decided to study the effect of adjusting the design parameters (e.g., window size, offset value, and frequency sub-bands) that affect the accuracy. Similarly, we combined this exploration with the selection of the best classifier providing the best accuracy. In fact, the modification of the above-mentioned parameters is commonly used in BCI research. In our validation, we used short trials in both the training and test sets, because some subjects cannot generate useful information along the full trial time. Such a procedure can be achieved by specifying the start and the end of the trial using the offset and time interval of the trial. Table 3 shows the results obtained by 121 iterations for different offset values and for different time window sizes. We started by fixing the time window and executing the algorithm with 11 iterations for all offset values from −0.5 to 0.5 with a step increase of 0.1. Then, we changed the time window and proceeded with the same accuracy calculations for all possible offset values, as depicted in Table 3. The best result was obtained for the 2-second window size with a zero-value offset, where the duration started in the third second and ended in the fifth second of the trial. The window size and the offset value were customized during the training phase to be fixed for the remaining testing phase, and both were related to the recording conditions of EEG signals. This approach was applied for both datasets IIa and IVa provided by the BCI competition. The results obtained with the IIa dataset were more significant, because it integrated three times more subjects than those presented in the IVa dataset, as shown in Table 3. The same approach can be adapted during the training phase for other datasets to find the best offset and time window values.

The accuracy decreased with the increase or decrease of the size of the trial window. Thus, the high value of the offset to the right led to a high accuracy close to 84%. This value was probably reached because some subjects started imagining before the stimulus, which is considered an error during EEG data recording. However, the accuracy decreased when the offset value was greater than zero, meaning that ERD information is directly located after starting the trial for most subjects. Consequently, taking a high offset around the trial causes a loss of useful information. As a result, the best classification accuracy was 85.12% for the 2-second time window without any offset shifting.

In this step, the best values of offset and time window size obtained from the previous step were used. The frequency bands (8–32 Hz) when the amplitude and phase were extracted using FFT were modified to determine the most convenient frequency bands to maximize the accuracy for the current dataset. As in the customization of time window and offset values, the frequency bands were adjusted with respect to the FFT and AR features to select the best two subbands, which were then fixed for the remaining testing phase. Such modifications were just related to the FFT features, while both frequency bands of AR remained constant (8–18, 18–32 Hz). Fourteen frequency bands were tested, and the results are presented in Table 4. We noticed that the best results were obtained with the 8–34-Hz frequency band. Most frequency bands, especially those with upper stop bands, presented results lower than 32 Hz. Poor results were obtained using the 7–24-Hz frequency band, as depicted in Figure 3a.

Similarly, we proceeded by modifying the frequency bands of the AR algorithm, as mentioned in Figure 3b, to examine their effect on the total accuracy. Apparently, the highest accuracy was obtained for the 8–15-Hz frequency band with a small increase of 0.2 compared with the accuracy computed with the 8–34-Hz frequency band used in the FFT technique. As with the KNN classifier, we proceeded to check our design with the other classifiers (i.e., the multi-LDA and multi-SVM classifiers). Each multi-classifier was built with three two-class classifiers, as presented above. A voting technique was added to each multi-class classifier to provide the best classification results. As shown in Table 4, the results obtained from the multi-LDA and multi-SVM classifiers were better than those calculated using the KNN classifier. The results of the multi-SVM were very close to those obtained by the multi-LDA, but the multi-LDA gave about 1% better accuracy. Moreover, the multi-SVM took more time to learn how it classified the features in the training step, but this time delay is not important in the case of training (see Table 5). In this case, the multi-SVM took a time average of 0.734 s, while the multi-LDA took only 0.0164 s, which was even faster than the KNN classifier (0.033 s).

Feature selection was also integrated to assist the LDA classifier to solve the problems of high dimensionality by reducing its feature vector. Table 6 shows the number of features selected by feature selection for each block of classifier as a combination. All classifier blocks in the multi-LDA had almost the same efficiency in the separation and classification of the features (i.e., accuracy of about 50%–60%).

Table 7a,b shows the number of true decisions for the multi-LDA and classification rates, respectively. The accuracy was low, because each module could only classify two classes (66% of the total trials), but the third class was incorrectly classified. That means the accuracy for each module was over 90%.

All the results presented above were conducted on dataset IIa. We completed our validation on the other dataset (i.e., IVa). Table 8 presents the results obtained from dataset IVa. The results of this dataset were better than those of dataset IIa, especially for the first subject that reached an accuracy level of 100%.

From the experiment, we found that the best approach was the algorithm that combined features of the amplitude and phase with features extracted by AR modeling of the reconstructed phase-space features. This method was applied to 15 EEG channels filtered by Laplacian surface and band-pass filters. Through feature selection, the optimal features for multi-LDA were selected. The final results are shown in Table 9. We noticed that the algorithm was able to efficiently classify both datasets. The final approach is described in Figure 4.

## 4. Discussion and Conclusions

This paper has provided an exhaustive design exploration and analysis of three-class MI classification through the integration of a sophisticated feature extraction technique. The proposed method combines the features of the phase and amplitude of the brain signals using FFT and AR modeling of the reconstructed phase space. The classification is conducted with multi-LDA and multi-SVM classifiers where a voting technique is integrated to provide a robust result. An exhaustive exploration of some design parameters (e.g., offset value, window size, sub-band frequency range) has been conducted during the training phase to customize our design to maximize the classification accuracy. Our results show that reliable feature extraction combined with well-adapted filtering techniques can be obtained for three-class motor imagery. The classification rates achieved after customizing all design parameters (e.g., window size, sub-bands, offset) were relatively good. Feature computation followed by classification is extremely fast, suggesting that this approach is appropriate for real-time BCI and other online applications.

Table 10 shows a comparison of our results and the published results. The BCI competition was based on kappa for validation. Therefore, we also showed the results based on kappa to simplify the comparison. Most previous research presented four-class BCIs, whereas we presented a three-class BCI. Therefore, the comparison between our results and the published results is accurate, but it is clear that the results are better than those expected if the approach of the BCI competition were only used for three classes. Most subjects in both datasets had presented more results than those mentioned in the literature review.

The winner of dataset IIa was Kai, who proposed a filter bank common spatial pattern (FBCSP) method for eight frequency bands in 7–30 Hz for a four-class BCI. The classifier was based on a one-vs-rest strategy using Nave Bayes. Heung used the same data, but it was only for five subjects. They used a combination between CSP and wavelet coefficients. They calculated the classification accuracy, but we converted the results into kappa values. The winner in the BCI competition (Guan) used multi-class CSP and SVM and feature selection, but only for dataset IVa, and reached 0.792 for four classes. Townsend compared CSP and BP containing the amplitude and phase and found that BP was better than CSP.

To complete the proposed classification system evaluation, the Information Transfer Rate (ITR) has been measured for all EEG signal processing of the system during the offline validation process. The ITR can be expressed as in [39]: (11)$$ITR=L\times [p\times \log_{2}(p)+\log_{2}(N)+(1-p)\times \log_{2}(\frac{1-p}{N-1})]$$ where *L* is the number of decisions per minute, and *p* the accuracy of the decision made for the *N* targets. A set of metrics are used to evaluate the performance of our proposed system as well as similar architectures based on P300, SSVEP and motion paradigms, including: information transfer rate (ITR), and scores incremented by one for when the symbol selected by the BCI matched the target symbol. As depicted in Table 11, our measurements provide about 93.33% for the accuracy of twelve subjects with ITR close to 21 bits/min, which are sufficient for motor imagery [40].

As depicted is Table 12, It has been reported in many studies that different couples of feature extraction and classification techniques can be used in three-class based motor imagery with an accuracy varying from 60% to 90%. Our proposed work presents reasonable results with an accuracy of 86% to 93%.

As depicted in Figure 5, the receiver operating characteristic (ROC) curves provide a plot of the sensitivity against its specificity applied on IVa data set. The area under the curve is a very informative statistic for the evaluation of the performance of the classification algorithm as well as for the analysis of the usefulness of its features. Even if the multi-LDA ROC curves provide an area under the curve better than that of the multi-SVM classifier, the difference between them remains limited for all subjects.

The Figure 6 presents the ROC analysis for the best classifier applied for the IIa Data set. It is worth noting that the accuracy is not the same for all subjects. For example for subjects 1, 2 and 3 the results are almost the same where for subjects 4,5 and 6 the results give a performance slightly worse that the remaining subjects. We estimate that this difference is due to the impairment on emotional and social cognition that could be also a problem to affect the executive function in BCI practice [43]. Furthermore, the emotion changes in each subject were not assessable during the EEG signal acquisition. On the other hand, the intrinsic characteristic if the EEG signals containing unwanted artifacts, which could lead to experiment degradation [44]. However, the results averaged over all nine subjects remain interesting with 86%.

We thoroughly analyzed several ensemble methods applied to three-class classification problems. Therefore, we have tried to provide a design that gives us the best classification rate for different real-life datasets by: Applying the phase-space reconstruction techniques with the AR model for the feature extraction part;Implementing the three-class classifier based on the one-against-one strategy for both the multi-LDA and multi-SVM classifiers;Customizing all design parameters during the training phase to reap the benefit of their effect on the classification accuracy;Considering the cross-validation approach to reduce the effect of the unbalanced data that characterize the EEG signals.


The experimental results have shown that the LDA classifier combined with state-space reconstruction techniques for feature extraction provided better classification results than SVM and KNN. Extracting features by amplitude–phase analysis combined with AR achieved better results than the classification performance of the BCI competition (2003 and 2008) Graz datasets. The phase-space reconstruction method increased the performance of the system more than those winning methods in the Graz 2008 dataset. Since the SFFS has just been used in training, it is not considered a drawback of the system. Furthermore, the timing analysis of the proposed architecture indicates that we can implement our design in real-time applications. It is also expected to be appropriate to be applied as an embedded system design with sufficient performance. In future work, we will study this approach using field programmable gate array (FPGA) technology, and we will try to extend it to four classes and check its performance.

Furthermore, we evaluated the cost using an FPGA-based platform of Altera incorporating a fast version of the Nios-II core processor with on-chip memories and appropriate interfaces to interconnect co-processors to the Avalon bus. This interface is exclusively used to build all Altera system designs to simplify the interconnection and to manage the communication within a complex architecture including a multiprocessor organization. The execution time of each part was calculated on the Nios-II core processor at the frequency of 150 MHz. As illustrated in Figure 7, the execution time for the feature extraction is longer than that for the classification, because the feature extraction contains many algorithms, such as FFT and AR models. This delay is acceptable and does not constitute a problem in our system. Likewise, the high time delay of the feature selection is not important, because it is only applied in the case of training. The high time delay of the preprocessing section makes it a critical part of the system in the case of testing (0.55 s). Hence, the system takes a total execution time of 0.93 s to decide LH, RH, or F class, and so, this delay should be reduced by implementing the filter block in the hardware design as a co-processor using Verilog language.

A line for improvement is to incorporate strategies for a continuous adaptation of the feature extraction algorithm to account for the non-stationary characteristics of the EEG. Furthermore, the integration of the proposed architecture as an embedded system running on soft core processor and operating in real-time will be a promising extension and is subject of present research in our team.

## Figures and Tables

**Figure 1 brainsci-06-00036-f001:**
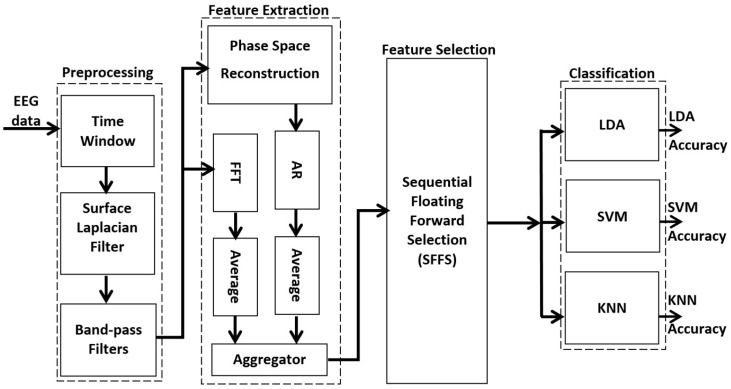
Proposed framework for MI-BCI multi-class classification.

**Figure 2 brainsci-06-00036-f002:**
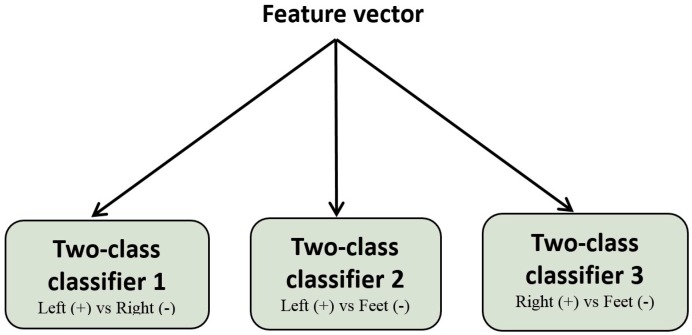
One-Against-One strategy based three-class SVM and LDA classifiers (multi-SVM, multi-LDA).

**Figure 3 brainsci-06-00036-f003:**
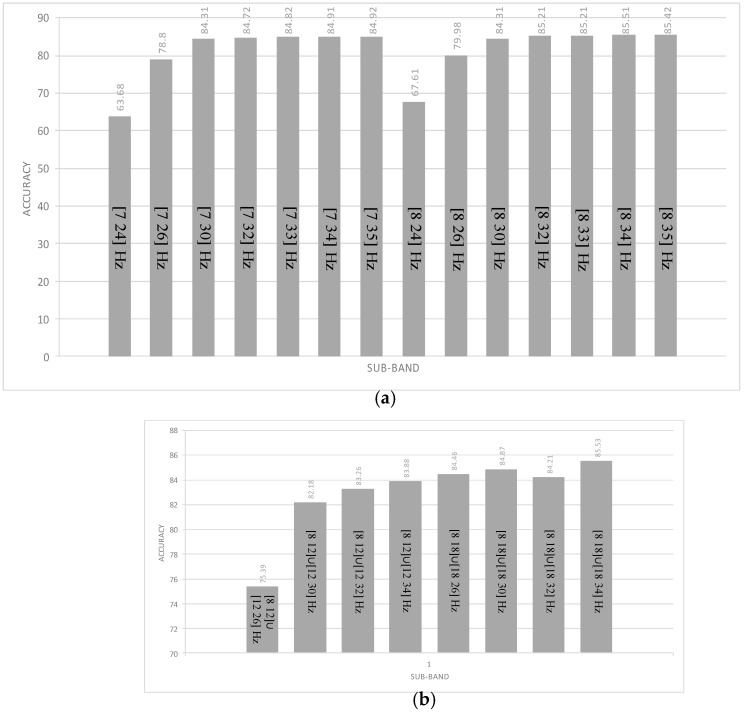
The effects of frequency bands on the classification accuracy. (**a**) Effect of the offset and time window variation on the accuracy using fast Fourier transform (FFT); (**b**) Effect of the frequency band variation on the on the accuracy using autoregressive (AR) model.

**Figure 4 brainsci-06-00036-f004:**
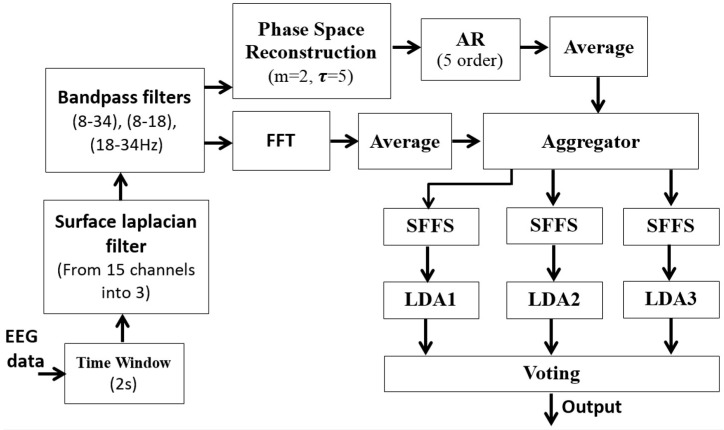
The best approach of three-class BCI system.

**Figure 5 brainsci-06-00036-f005:**
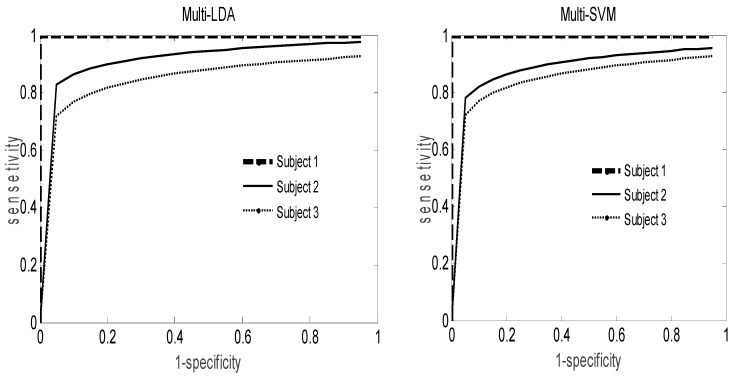
ROC curve analysis for the proposed classifiers.

**Figure 6 brainsci-06-00036-f006:**
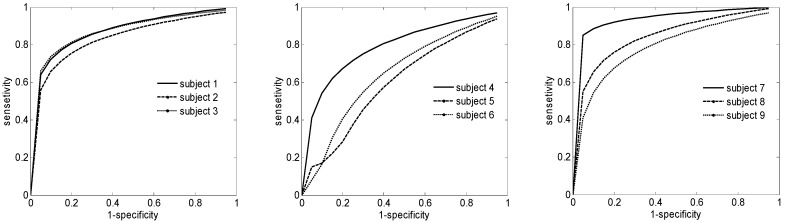
ROC curve analysis for IIa Data set using the best classification results.

**Figure 7 brainsci-06-00036-f007:**
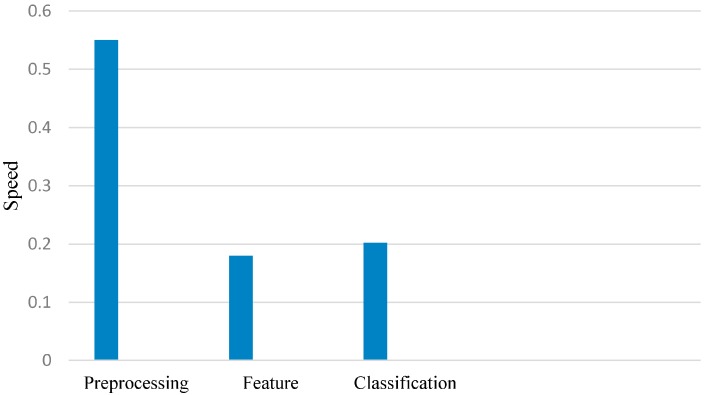
The execution time of each part for the proposed algorithm on FPGA.

**Table 1 brainsci-06-00036-t001:** Voting method for three MI states.

Classifier 1 (LH vs. RH)	Classifier 2 (LH vs. Feet)	Classifier 3 (RH vs. Feet)	Vote Output
LH	LH	∀	LH
RH	∀	RH	RH
∀	Feet	Feet	Feet
LH	Feet	RH	None
RH	LH	Feet	None

LH: Left Hand, RH: Right Hand and ∀: whatever LH or RH.

**Table 2 brainsci-06-00036-t002:** The classification accuracy for each subject (Si) of the IIa data set.

	S1	S2	S3	S4	S5	S6	S7	S8	S9	Mean
Accuracy (%)	92.59	85.18	91.66	75	65.2	63.42	91.66	89.81	71.75	80.71

**Table 3 brainsci-06-00036-t003:** The average classification accuracy for 121 iterations (offset + time window).

	Time Window (s)											
		3	2.8	2.6	2.4	2.2	2	1.8	1.6	1.4	1.2	1
**Offset amount**	**−0.5**	83.95	84.2	84.03	84.92	84.36	84.97	84.82	83.58	80.04	80.09	70.37
	**−0.4**	83.59	84.25	84.1	84.72	83.64	84.23	84.51	84.46	83.79	83.69	80.4
	**−0.3**	83.02	84.36	84.49	84.13	83.64	84.97	82.61	84.87	84.03	84.25	84.2
	**−0.2**	82.66	84.97	83.23	83.33	83.53	83.49	84.2	84.61	84.28	84.2	82.03
	**−0.1**	81.43	84.05	84.92	84.97	84.1	83.18	84.31	84.08	84.01	84	83.87
	**0**	80.70	83.12	84.77	84.82	84.46	85.21	83.15	80.04	65.89	62.19	59.36
	**0.1**	78.80	79.83	83.53	84.82	84.31	79.16	59.56	58.95	67.84	55.70	58.33
	**0.2**	75.87	77.31	83.17	84.36	82.97	80.70	62.34	63.68	60.08	61.52	58.38
	**0.3**	72.17	73.50	81.43	82.76	82.09	81.36	82.35	61.93	59.82	56.79	63.11
	**0.4**	67.74	69.65	79.16	81.37	82.97	83.69	83.74	58.64	61.41	55.29	67.43
	**0.5**	62.70	64.76	75.30	78.60	82.51	81.43	84.61	72.17	60.95	57.56	60.18

**Table 4 brainsci-06-00036-t004:** Multi-class LDA and SVM classification accuracy results.

	Classification Results	
Subject	Multi-LDA	Multi-SVM
1	92.13	90.74
2	89.352	89.352
3	92.13	93.519
4	86.111	86.574
5	66.667	68.056
6	74.537	69.444
7	96.759	95.833
8	90.27	89.815
9	86.57	86.574
Mean	86.06	85.545

**Table 5 brainsci-06-00036-t005:** Processing time for multi-LDA, multi-SVM and KNN during the training phase.

	Processing Time (s)		
Subject	Multi-LDA	Multi-SVM	KNN
1	0.0134	0.526	0.029
2	0.0177	0.739	0.053
3	0.038	0.279	0.0248
4	0.0113	0.794	0.0283
5	0.0148	1.44	0.0312
6	0.0127	0.422	0.027
7	0.0140	0.416	0.0319
8	0.0169	0.96	0.0266
9	0.090	0.896	0.050
Mean	0.0164	0.734	0.033

**Table 6 brainsci-06-00036-t006:** The number of features selected using SFFS.

	Number of Feature Selected by SFFS		
Subject	LH & RH	LH & F	RH & FH
1	10	4	3
2	10	5	6
3	5	4	3
4	11	2	4
5	16	21	21
6	15	11	16
7	3	5	11
8	10	8	5
9	9	5	6

LH: left hand, RH: right hand.

**Table 7 brainsci-06-00036-t007:** Multi-LDA classification results true decision number

	Number of Feature Selected by SFFS		
Subject	LH & RH	LH & F	RH & FH
**(a) Multi-LDA True Decision Number**			
1	134	140	141
2	132	134	138
3	137	137	135
4	125	129	131
5	115	118	109
6	143	123	119
7	143	138	144
8	136	132	144
9	132	131	130
**(b) Multi-LDA Classification Rate (%)**			
1	62.037	64.815	65.27
2	61.111	62.037	63.88
3	63.42	63.42	62.5
4	57.87	59.72	60.64
5	50	54.63	50.46
6	53.24	56.94	55.09
7	66.204	63.88	66.66
8	62.96	61.11	64.35
9	61.11	60.6	60.18

**Table 8 brainsci-06-00036-t008:** Classification accuracy results with multi-LDA and multi-SVM.

	Classification Results	
Subject	Multi-LDA	Multi-SVM
1	100	100
2	93.3	90
3	86.7	86.7
Mean	0.93.33	92.23

**Table 9 brainsci-06-00036-t009:** The final classification results.

Frequency Band	Time	Number of Channels	Classifier	Cross Validation	Classification Accuracy Average
8–34 Hz	2 s	15	Multi-LDA	9 folds	86.06% for Iia; 93.3% for IVa

**Table 10 brainsci-06-00036-t010:** A comparison between BCI competition and our results.

**(a) Data Set (IIa) Based Results**			
	**Contributor of Data Set (IIa)**		
**Subject**	**Ken [36]**	**Suk [37]**	**Our Result**
1	0.68	0.69	0.88
2	0.42	0.92	0.84
3	0.75	0.36	0.88
4	0.48	0.88	0.79
5	0.40	0.30	0.50
6	0.27		0.61
7	0.77		0.95
8	0.75		0.85
9	0.61		0.79
Mean	0.57	0.63	0.78
**(b) Data Set (IVa) Based Results**			
	**Contributor of Data Set (IVa)**		
**Subject**	**Guan [38]**	**Tow [16]**	**Our Result**
1	0.755	0.9	1
2	0.800	0.78	0.90
3	0.792	0.83	0.801
Mean	0.792	0.836	0.90

**Table 11 brainsci-06-00036-t011:** ITR comparison of different application in BCI competition [40].

Team	System	Type	Paradigm	P (%)	T (sec/syn)	Score	ITR (bits/min)
1	Biosemi	Synchronous	Motion	82	7.2	32	30.8
2	TsinghuaMiPower	Synchronous	SSVEP	87.88	10.9	25	23.8
3	G-Tec	Synchronous	SSVEP	55.32	7.66	5	15.4
4	g.USBamp [22]	Asynchronous	MI	92.17	2	32	18.11
5	Our Architecture	Synchronous	MI	93	2	36	21

ITR: information transfer rate, SSVEP: steady state virtually evoked potential, MI: motor imagery.

**Table 12 brainsci-06-00036-t012:** Classification accuracy of three-class based motor imagery.

Team	Classes State	Feature Extraction	Classifier	Accuracy (%)
1 [3]	LH, RH, Word	FFT	KNN	80
2 [41]	LH, RH, Feet	PSD + GA	LDA (one vs. one)	60–90
3 [5]	LH, RH, Word	SL Filter + PSD + SFFS	LDCRF	69.5
4 [42]	LH, RH, Feet	Wavelet + CSP + FDA	SVM	88
5 [11]	LH, RH, Feet	Data Interception + BP features + CSP	LDA	85

GA: genetic algorithm; LDCRF: latent dynamic conditional random fields; SL Filter: spherical surface Laplacian; CSP: common spatial pattern; PSD: power spectral density; FDA: Fisher discriminant analysis; SFFS: sequential forward floating selection; BP: band Power.

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
