# Peer review of "Three-Class EEG-Based Motor Imagery Classification Using Phase-Space Reconstruction Technique"

_brainsci, 2016, doi:10.3390/brainsci6030036_

Round 1
Reviewer 1 Report
The presented work presented focus on a BCI application and in particular, classifying 3 types of motor imagery ERP tasks including left hand, right hand and foot movements using machine learning techniques. A combination of time-, frequency- and phase-domain features followed by a feature selection was utilised to distinguish the 3 mentioned tasks using 3 classic LDA, SVM, and KNN classifiers. In my view, the work has some values and hence it deserves to be published in the Journal of Brain Sciences. However, there exist some structural and technical issues, which need to be addressed before publication. More comments are provided below.
General comments
· Please use small words for abbreviations, for example, fast Fourier transform in abstract, surface Laplacian filter in L142P4 (L=line, P=Page), autoregression in L93P2, one-against-one in L103P3, left hand in L133P4.
· Please avoid unnecessary abbreviations e.g., LDCRF in L43P1, LDCRF in L43P1, CRF in L44P1. Please be consistent in using e.g. K-NN (or KNN).
· I came across many typos throughout the manuscript e.g. ‘distinguishing’ in L75P2 and ‘convert’ in L93P2, which I do not intend to address all at this stage, but please correct them for the next round.
· When citing a paper within the text use et al. (referring to a number of people), for example, Townsend et al. in L85P2 (L=line, P=Page). Please double check the name of the authors. For example, the author of [15] is Feng et al. (not Yonghui) in L83P2.
· Please avoid using “etc.” and “…”, “would like to”, and “So” (use hence, therefore, accordingly or hereupon instead).
Affiliation
· The numbers and email addresses are not consistent with authors’ names.
Abstract
· ERD/ERS does not require abbreviation. Similarly, for BCI, FFT, and SFFS. Abstract is an independent document. Please use an abbreviation if it is repeated within the abstract.
· What do you mean by satisfactory? With respect to the results of other studies or in compare with single-modal feature analysis? Please be clear.
· Please add a take home message to the end of the abstract.
Keywords
· Electroencephalogram (EEG)
Introduction
· (MI)-based in L31P1
· Pls revise ‘Since the EEG …’ sentence in L39P1.
· Cite SFFS in L42P1. It seems sequential forward floating selection algorithm is the correct name for SFFS. Is it?
· Pls revise ‘Quadratic discriminant …’ sentence in L47P2.
· Pls revise ‘The complexity …’ sentence in L55P2
· Pls revise ‘Nevertheless, CSP is …’ sentence in L62P2
· Pls revise ‘By this method …’ sentence in L93P2
· L107P3: no need to introduce the contents of the next (sub)sections.
· The objectives and motivation of the work should be clearly mentioned in the introduction section.
Materials and Methods
· Pls revise ‘We modified …’ sentence in L93P2.
· Pls revise ‘Both SVM …’ sentence in L122P3.
· L132P4, ‘were performed’ and not ‘were trying to perform’.
Figures/table
· Captions require more details.
Preprocessing
· L156P4, Cite Pfurtscheller and Lopes da Silva, (1999) for ERD/ERS.
· Pls revise ‘We used …’ sentence in L156P4. Sharper is what sense? And, with respect to which another filter type?
Feature extraction
· ‘To enhance the analysis’ instead of ‘In order to get the maximum contributive features’ in L168P4.
· [32] and not [32} in L178P5.
· Short ‘enough’ in L180P5.
· ‘a’ single vector in L193P5.
The combined Phase-space reconstruction and autoregressive model technique
· Pls revise ‘The phase-space reconstruction …’ sentence in L195P5.
· Eq (3), xi and not ‘Xi’ and,
· m is the dimension of what? L212P5.
· Did the authors validate the AR model and the coefficients e.g. by Durbin-Watson whiteness test (Durbin and Watson 1951) or the consistency check of the correlation structure of the Ding method (Ding et al. 2000).
Feature selection
· It is not clear how the SFFS selects the optimal features. This should be briefly described. Besides, how did the author validate the optimal features? Perhaps, statistical tests e.g. one-way ANOVA could be used for e.g. check whether the selected features across 3 groups are statistically significant.
Classification
· Table 1 and its above explanations are quite confusing. Why there is no + or - in the first cell of the table? Please make it clear by adding an explanation to the caption.
· Why didn’t you use a multi-class classification technique e.g. Gaussian process classification (Seeger, 2004)? Perhaps, this can be discussed in the discussion section.
Results and discussions
· I would separate results from discussions.
· It is not clear why the SVM classifier provided a lower classification accuracy than the LDA. In most cases, the non-linear SVM method is superior to the linear LDA. Perhaps, this can be further elaborated.
· It is not clear why only 3 frequency bands (8-34, 8-18 and 18-34) were selected for the analysis.
· It seems Tables 6-8 are not helpful (just a suggestion!).
· I would report the sensitivity and specificity of the classification (an ROC plot) results in Table 9 too.
Conclusions
· Perhaps conclusion can be merged with the discussion section after the revision. The discussion of the work is weak. Please improve it for the next round.
· Pls add the full name of FPGA.
Abbreviations
· Is this part necessary?
(My) References
Pfurtscheller, G., Lopes da Silva, F.H., 1999. Event-related EEG/MEG synchronization and desynchronization: basic principles. Clin. Neurophysiol. 110, 1842–57.
Seeger, M., 2004. Gaussian processes for machine learning., International journal of neural systems. doi:10.1142/S0129065704001899
I hope my comments would be helpful.

Author Response
The authors would like to thanks the reviewers for their valuable comments and we present the author’s responses in the file attached.

Reviewer 2 Report
In this paper entitled “Three-class EEG-based motor imagery classification using phase-space reconstruction technique”, the authors present a method for combining the features of the phase and amplitude of the brain signals using Fast Fourier Transform and autoregressive modelling of the reconstructed phase space, as well as the modification of the BCI.The introduction is well written and it gives the main methods of the state of the art.
Table 1 is not clear, the authors should better explain how the results are combined.
Page 5: “The classification accuracy ….” It seems a part is missing.
Table 3, the authors should mention what parameters provide a significant difference across subjects.
Table 4 would work better as a figure. Again it is worthwhile to know what parameters provide a significant impact.
Tables in table 8 could be merged and better presented.
As opposed to other people in the literature who use one session to train and another to test, the authors use a cross validation procedure for the evaluation, which does not represent what happens with a real BCI. The evaluation, and the comparisons with other works, should be clarified.
The results and the discussion section should be separated.
Author Response

(The authors gave the same response as above.)

Round 2
Reviewer 1 Report
The current revision shows an effective improvement, however, there still exist some issues that need be addressed before publication. Specific comments are given in bellow:
Feature selection,
o The description of the SFFS algorithm is not clear. How the best combination of candidates are selected/evaluated? Based on (10-fold) cross-validation?
o It seems the SFFS can be computationally costly, especially for real-time BCI applications. An alternative would be SFS [Pohjalainen et al., 2015]. Perhaps, this can be elaborated in the dissuasion section.
Classification,
o Following my previous query, I have not completely convinced the LDA classifier results in a better accuracy than the SVM. As far as I know, an SVM with linear kernel can be as good as LDA.
o A balanced accuracy or BA (average of sensitivity and specificity) would be a better measure to report the classification accuracy.
· Data description
o The description of the data used in this study is not clear. It seems, both dataset IIa [Blanchard and Blankertz, 2004] and IIIa [Blankertz et al., 2006] contain 3 subjects. Please devote one subsection to the Data description.
Minor issues, (please proofread the whole manuscript for the next round)
· The first sentence is not necessary. The objectives, motivation, and innovation of the work is not clear. For example, how this work stand against the other similar studies? Perhaps, you should focus on Phase-space reconstruction technique as the innovation of the work, consistent with the title.
· Here and there, I found unclear sentences, e.g.
o L13P1, ‘The classification approach …’ (SFFS can be abstract )
o L16P1, ‘We reported interesting …’
· Full name of some abbreviations has been repeated throughout the manuscript, e.g. SFFS, LDA. I suggest using ‘consistency checker’ that is a free MS word app to correct the mentioned issues.
· Sorting of affiliations is not correct.
· Fig. 6 is a bit stretched and is not clear. It seems it shows the results from only one dataset. How about the other? Please correct the x-label ‘1-specificity’.
References:
Blanchard G, Blankertz B (2004): BCI competition 2003 - Data set IIa: Spatial patterns of self-controlled brain rhythm modulations. IEEE Trans Biomed Eng 51:1062–1066.
Blankertz B, Muller KR, Krusienski DJ, Schalk G, Wolpaw JR, Schlogl A, Pfurtscheller G, Milllan JDR, Schrder M, Birbaumer N (2006): The BCI competition III: Validating alternative approaches to actual BCI problems. IEEE Trans Neural Syst Rehabil Eng 14:153–159.
Pohjalainen J, Rasanen O, Kadioglu S (2015): Feature selection methods and their combinations in high-dimensional classification of speaker likability, intelligibility and personality traits. Comput Speech Lang 29:145–171.

Reviewer 2 Report
The authors have done some substantial changes in the paper, and it has improved its quality. Overall, it would be worthwhile if the authors could give the paper to a native English speaker to check and update the style, as it would definitively help the reader to go through the different sections.
Major issue: there is no statistical analysis with comparisons between the different methods, and the evaluation is performed with a cross validation procedure, it doesnt represent how a BCI will be applied, authors should clearly separate training from the test, in order to compare with other published results in the literature.
Figure 2 is not a figure but an algorithm. To find a "goto" in an algorithm is not acceptable, the authors should redefine the algorithm with a "while" structure.
"Previous researches" -> "Previous studies"
"So, the length of FFT used was chosen with 64 samples" -> "The length of the segment used by the FFT was set to 64 samples"
Why table 2 does not include the results of SVM and LDA?
Page 16 (on top of table 3): "was obtained for the data sets IIa and IIa", it is confusing.
Table 10: dataset 2a has 9 subjects, why only three subjects are presented? Are the tables inverted between 2a and 3a?
The size of the font is not consistent in the sections (table and figure captions). The authors should respect the layout of the journal. It would be better to split discussion and conclusion.
The references should have the same style throughout the reference section. The authors could include other recent work using the same databases:
(1) H. Raza, H. Cecotti, Y. Li, G. Prasad, Adaptive Learning with
Covariate Shift-Detection for Motor Imagery based Brain-Computer
Interface, Soft Computing. 2015
(2) H. Raza, H. Cecotti, G. Prasad, Optimising Frequency Band
Selection with Forward-Addition and Backward-Elimination Algorithms in
EEG-based Brain-Computer Interfaces, International Joint Conference on
Neural Networks. 2015
Round 3
Reviewer 1 Report
I would like to appreciate all efforts by the authors, the manuscript has improved effectively as compared with the previous version. I am now convinced by the answers, but I am afraid the issue with the English still remains. I suggest asking a professional (a senior researcher) to help you on this matter. In addition, some of my previous comments have not been addressed properly. Please try to address them for the next round of review.
Author Response
Dear respectable reviewer,
The authors would like to thanks the reviewers for their valuable comments elaborated during the third round and present responses in the file attached.
Kind regards,
The corresponding author.

Reviewer 2 Report
The authors have made some substantial changes in the paper. However, there are still some issues.
There is an issue with the place of + and - of x in the algorithm, it s unclear what is the meaning of the position of the sign, and if it is correct.
Again, it should be noted as an algorithm, and not a figure.
"We estimate that this 523 difference is due to the emotional state of the subject and the non stationary character of the acquired EEG signals." : by what evidence is it supported?
There is no statistical analysis with other reported results in the literature or a baseline method, it is a key issue to assess the relevance of the proposed method.
Table 11. It is not clear how the authors have presented a new amplifier? There is a problem in the description of the table. In addition, what is the reference/description of the other systems? The authors did use cross validation with offline data, therefore it does not simulate what is a real BCI.
The style of the reference is still not consistent, as the first few references are smaller.
Author Response
Dear respectable reviewer.
The authors would like to thanks the reviewers for their valuable comments elaborated during the third round and present responses in the file attached.

Round 4
Reviewer 1 Report
I believe the manuscript is currently sufficient for the publication although after some minor corrections. Please see some suggestions that I came across in below:
L16, By utilizing a sequential forward floating selection (SFFS) and a multi-class linear discriminant analysis (LDA), our findings showed superior classification results, a classification accuracy of 86.06% and 93% for two BCI competition datasets, with respect to results from previous studies.
L36, several citations are required. E.g., three-class ... have been developed [3-10].
L66, several citations are required.
L 107, ..., SVM)
L 116, based on MATLAB simulation
L130, (K-nearest-neighbor, KNN)
Fig. 1, caption: Proposed framework for MI-BCI multi-class classification.
L 211, ...lower in compare with ..
L 234, can be considered as ..
L 222, M = N-(m-1) should be defined in the beginning of the sentence.
L 294, 10-fold ..
L 336, please remove 'in this section'.
L399, as presented above.
Author Response
Dear respectable reviewer,
Please find attached the authors responses for your comments.
Many thanks for your effort and for your interesting comments.
Kind regards,
Dr. Ridha Djemal
